# Atomic structure of the 26S proteasome lid reveals the mechanism of deubiquitinase inhibition

Corey M Dambacher[1†], Evan J Worden[2,3†], Mark A Herzik Jr.[1†], Andreas Martin[2,3,4*], Gabriel C Lander[1*]

[1]Department of Integrative Structural and Computational Biology, The Scripps Research Institute, La Jolla, United States; [2]Department of Molecular and Cell Biology, University of California, Berkeley, Berkeley, United States; [3]QB3 Institute, University of California, Berkeley, Berkeley, United States; [4]Howard Hughes Medical Institute, University of California, Berkeley, Berkeley, United States

**Abstract** The 26S proteasome is responsible for the selective, ATP-dependent degradation of polyubiquitinated cellular proteins. Removal of ubiquitin chains from targeted substrates at the proteasome is a prerequisite for substrate processing and is accomplished by Rpn11, a deubiquitinase within the 'lid' sub-complex. Prior to the lid's incorporation into the proteasome, Rpn11 deubiquitinase activity is inhibited to prevent unwarranted deubiquitination of polyubiquitinated proteins. Here we present the atomic model of the isolated lid sub-complex, as determined by cryo-electron microscopy at 3.5 Å resolution, revealing how Rpn11 is inhibited through its interaction with a neighboring lid subunit, Rpn5. Through mutagenesis of specific residues, we describe the network of interactions that are required to stabilize this inhibited state. These results provide significant insight into the intricate mechanisms of proteasome assembly, outlining the substantial conformational rearrangements that occur during incorporation of the lid into the 26S holoenzyme, which ultimately activates the deubiquitinase for substrate degradation.

*For correspondence: a.martin@berkeley.edu (AM); glander@scripps.edu (GCL)

†These authors contributed equally to this work

Competing interests: The authors declare that no competing interests exist.

## Introduction

The eukaryotic 26S proteasome is a large multi-enzyme complex that functions as the primary degradation machinery for the selective turnover of aberrant or unneeded proteins within the cell. Proteins targeted for degradation are covalently labeled with polyubiquitin chains, which are recognized and removed by the proteasome (*Finley, 2009*). The barrel-shaped core peptidase complex of the proteasome, which sequesters the proteolytic active sites in an internal chamber (*Groll et al., 1997*), is capped on one or both ends by a regulatory particle that acts as a discriminating gateway for targeted protein substrates (*Saeki and Tanaka, 2012*). The regulatory particle consists of two sub-complexes, known as the 'base' and the 'lid' (*Glickman et al., 1998*). The base sub-complex contains the AAA+ ATPases Rpt1-Rpt6, which form a heterohexameric ring that drives the mechanical substrate unfolding and translocation of the unstructured polypeptides into the degradation chamber of the core peptidase. Docked on one side of the base is the lid subcomplex, which contains the deubiquitinating enzyme (DUB) Rpn11 that cleaves polyubiquitin chains from targeted substrates as an essential step in proteasomal substrate processing (*Boehringer et al., 2012*).

The lid is an asymmetric, ~370 kDa complex that consists of 9 unique subunits (Rpn3, 5, 6, 7, 8, 9, 11, 12, Sem1) and exhibits a characteristic hand-shaped organization similar to that of the eukaryotic initiation factor 3 (eIF3) and the COP9 signalosome (CSN) (*Lander et al., 2012*; *Lingaraju et al., 2014*; *des Georges et al., 2015*). At the center of the lid, six **P**roteasome-**C**SN-e**I**F3 (PCI)-domain

**eLife digest** The proteins contained within cells are constantly under scrutiny by a sophisticated "quality control" system that tags damaged or malfunctioning proteins with chains made up of a protein called ubiquitin. These ubiquitin chains serve as markers that target these toxic proteins for destruction by a molecular complex called the proteasome.

Removing ubiquitin chains from toxic proteins is a critical step in their degradation by the proteasome. This task is accomplished by an enzyme called a deubiquitinase, whose activity is tightly controlled. However, it was not clear how this enzyme is kept inactive before it is incorporated into the proteasome complex.

The deubiquitinase is part of a sub-complex called the "lid", which attaches to the side of the proteasome. Dambacher, Worden, Herzik et al. used electron microscopy to solve the structure of the lid complex in high detail – so that it was almost possible to view individual atoms. This revealed that the deubiquitinase was in a conformation that was very different from what had previously been observed in fully assembled proteasomes.

The structures revealed that within the lid complex, a complicated network of interactions causes the deubiquitinase to be encompassed by neighboring subunits. This prevents the enzyme from interacting with ubiquitin chains. Importantly, this network of interactions appears to be set on a hair-trigger, as mutations that disrupt these interactions cause the deubiquitinase to be activated. As the lid complex integrates into the proteasome, the lid undergoes large-scale structural rearrangements; Dambacher, Worden, Herzik et al. expect that these disrupt the interactions that maintain the deubiquitinase in an inhibited conformation.

Due to their ability to regulate the activity of the proteasome, deubiquitinases are becoming increasingly popular drug targets. Therefore, probing how they are activated in more detail will be of great importance to cell biologists and also contribute substantially to biomedical research.

containing subunits (Rpn3, 5, 6, 7, 9, 12) interact via their winged-helix motifs to form a horseshoe-shaped scaffold. The amino-terminal domains of these 6 subunits extend radially like fingers from the central PCI horseshoe. The essential deubiquitinase Rpn11 is positioned in the 'palm' of the hand-shaped lid. Rpn11 is an **M**pr1-**P**ad1 **N**-terminal (MPN)-domain containing metalloprotease of the **JA**B1/**M**PN/**M**OV34 (JAMM) family and forms a heterodimer with an enzymatically inactive MPN-subunit, Rpn8. With the exception of Sem1, a small 87-residue subunit located at the interface of the N-terminal domains of Rpn3 and Rpn7 (*Bohn et al., 2013*), all lid subunits contain conserved C-terminal helices that assemble into a large bundle positioned next to the MPN heterodimer of Rpn11/Rpn8 in the palm of the complex (*Beck et al., 2012*).

Previous crystallographic and biochemical studies have focused on the mechanism of Rpn11, which acts as a highly promiscuous DUB to remove ubiquitins from the wide variety of substrates during their translocation into the proteasome, likely by cleaving the isopeptide bond between the substrate lysine and the first ubiquitin moiety of the attached ubiquitin chain (*Worden et al., 2014*; *Pathare et al., 2014*). The Rpn11/Rpn8 heterodimer is active in isolation (*Worden et al., 2014*), but is significantly inhibited in the context of the lid sub-complex and regains robust DUB activity in the assembled 26S proteasome (*Verma et al., 2002*; *Yao and Cohen, 2002*). The isolated Rpn11/Rpn8 heterodimer is not present at considerable levels in the cell, whereas the presence of the lid and its assembly intermediates containing Rpn11/Rpn8 have been previously observed and characterized (*Tomko and Hochstrasser, 2011*). The inhibition of Rpn11 activity in the isolated lid and its assembly intermediates might therefore be important to prevent spurious deubiquitination of proteins in the cell, given the high promiscuity of this DUB. It has been suggested that interactions with Rpn5 are possibly involved in Rpn11 inhibition in the isolated lid (*Lander et al., 2012*), but the specifics of this regulation and the mechanism by which Rpn11 is activated upon incorporation into the holoenzyme remain elusive (*Verma et al., 2002*; *Lander et al., 2012*; *Yao and Cohen, 2002*).

Here, we present an atomic model of the isolated lid sub-complex of the yeast proteasome, as determined by cryo-electron microscopy (cryoEM) (*Figure 1A–C*, *Table 1*, *Figure 1—figures supplements 1–4*), revealing the molecular mechanism for direct inhibition of the DUB active site, as well

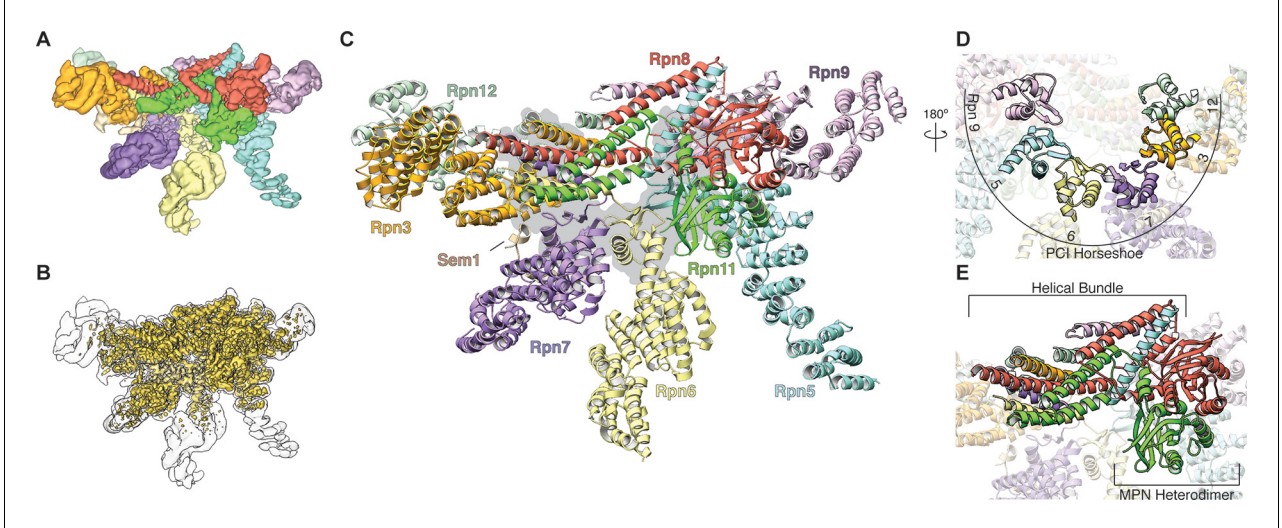

**Figure 1.** Architecture of the isolated proteasome lid sub-complex. (**A**) The segmented 3.5 Å resolution cryo-EM density is shown at a low isocontour level, with each subunit colored differently. Rpn3 is shown in orange, Rpn5 in light blue, Rpn6 in yellow, Rpn7 in purple, Rpn8 in red, Rpn9 in magenta, Rpn11 in green, Rpn12 in light green, and Sem1 in tan. This coloring scheme is maintained throughout all figures. (**B**) The unsegmented cryoEM density is shown at a higher isocontour level (in gold) to demonstrate the molecular details observable in the reconstruction (~3Å in certain regions). The lower isocontour level used for the segmented map is overlaid as a silhouette. (**C**) The atomic model of the proteasome lid is depicted using a ribbon representation, with each subunit colored according to the segmentation shown in A. The central location of the six PCI domains is illustrated by a gray shadow underneath the structure. (**D**) The PCI domains form a horseshoe, held together by an 18-stranded β-sheet. (**E**) Close-up of the helical bundle and the MPN heterodimer.

The following figure supplements are available for figure 1:

**Figure supplement 1.** CryoEM data collection.

**Figure supplement 2.** Single particle analysis of the lid complex.

**Figure supplement 3.** Resolution assessment of the reconstructions.

**Figure supplement 4.** Atomic modeling of the lid sub-complex.

**Figure supplement 5.** Subunits of the yeast 26S proteasome lid sub-complex.

**Figure supplement 6.** Comparison of PCI horseshoes in different complexes.

**Figure supplement 7.** Interactions of the helical bundle with surrounding lid components.

as Rpn11 activation through extensive conformational changes that occur during lid incorporation into the 26S holoenzyme.

## Results and discussion

### Lid architecture

Our cryo-EM reconstruction of the isolated lid shows that the MPN heterodimer, PCI horseshoe, and helical bundle together comprise a rigid substructure that contains regions resolved to ~3 Å resolution (*Figure 1B*, *Figure 1—figure supplements 3–5*). The N-terminal portions of the PCI-domain containing subunits progressively decrease in resolution as they extend toward the periphery of the complex, likely due to intrinsic flexibility (*Figure 1—figure supplements 3C*, *5*). The 3D reconstructions of the fully assembled lid and the lid lacking Rpn12 (*Figure 1—figure supplement 2*, top row, third reconstruction) show that the N-terminal portions of Rpn6 and Rpn5 are fully extended, and

**Table 1.** CryoEM data collection, processing, and modeling

**Data collection**

| | | | |
|---|---|---|---|
| | Microscope | | FEI Titan Krios |
| | Camera | | Gatan K2 Summit |
| | Voltage | | 300 keV |
| | Magnification | | 22,500 |
| | Pixel size | | 1.31 Å (0.655 Å, super-resolution) |
| | Dose rate | | 9.9 e⁻/pixel/s |
| | Cumulative electron dose | | 43.8 $e^-/Å^2$ |
| | Exposure | | 7.6 s |
| | Number of frames | | 38 |
| | Defocus range | | 1.5–3.5 μm |
| | Micrographs collected | | 3,432 |
| | Acquisition software | | Leginon (*Suloway et al., 2005*) |
| **Image Processing** | | | |
| | Preprocessing package | | Appion (*Lander et al., 2009*) |
| | Frame alignment software | | MotionCorr (whole image) (*Li et al., 2013*) |
| | CTF estimation software | | CTFFind3 (*Mindell and Grigorieff, 2003*) |
| | CTF cutoff criterion | | 4 Å at 0.5 confidence |
| | Particle picking software | | FindEM (*Roseman, 2004*) |
| | Micrographs used | | 3,365 |
| | Particles selected | | 254,112 |
| **Reconstruction** | | | |
| | Software | | RELION 1.3 (*Scheres, 2012*) |
| | Particles contributed | | 109,396 |
| | Rotational accuracy | | 1.392 degrees |
| | Translational accuracy | | 0.671 pixels |
| | B-factor applied | | -75.9 |
| | Final resolution @ FSC 0.143 | | 3.5 Å |
| **Model building and Refinement** | | | |
| | Number of residues | | 2743 (86%) |
| | Map CC (whole unit cell) | | 0.758 |
| | Map CC (all atoms) | | 0.853 |
| | R.M.S deviations | | |
| | | Bond length (Å) | 0.02 |
| | | Bond angle (°) | 1.15 |
| | Ramachandran plot stats | | |
| | | Preferred | 2646 (96.47%) |
| | | Allowed | 92 (3.35%) |
| | | Outlier | 5 (0.18%) |
| | Rotamer outliers | | 5 (0.20%) |
| | C-beta deviations | | 0 (0.00%) |

the MPN heterodimer and helical bundle adopt identical orientations in both structures. These findings contradict a recent crosslinking study (*Tomko et al., 2015*) suggesting that incorporation of

Rpn12 during lid assembly induces a large-scale rearrangement of the MPN dimer and the transition of the N-terminal portion of Rpn6 from an inward-folded state to an extended conformation that allows binding to the base. Further structural studies will therefore be required to better understand how Rpn12 incorporation affects lid binding to the base and core subcomplexes.

We found that the six PCI winged-helix domains associate into a continuous 18-stranded β-sheet, forming an incomplete right-handed spiral at the center of the lid sub-complex (*Figure 1C, D*, *Figure 1—figure supplement 6*). This organization was also observed in the crystal structure of CSN, although the PCI horseshoe assembly of the isolated lid has a wider and steeper spiral (*Figure 1—figure supplement 6*). Recently, a similar succession of β-strands was shown for the PCI domains in eIF3 (*des Georges et al., 2015*), but its domain organization is significantly more open and deviates from the spiral configuration observed in the proteasome lid and CSN (*Figure 1—figure supplement 6*). The significant conformational differences between the horseshoes of the lid, CSN, and eIF3 indicate that the PCI-domain assembly allows for substantial flexibility, while simultaneously serving as an organizational scaffold at the center of the complex.

The C-terminal helices of all lid subunits (except Sem1) assemble into a well-defined helical bundle that is centrally positioned within the lid sub-complex, adjacent to the PCI horseshoe and the MPN heterodimer (*Figure 1E*, *Figure 1—figure supplement 7*). Our cryoEM reconstruction contains sufficient structural detail to generate a complete atomic model of this helical bundle, providing an accurate depiction of the extensive inter-helical interactions. Furthermore, we were able to precisely assign the register of several helices that could not be unambiguously positioned in earlier lower-resolution models of this bundle (*Beck et al., 2012*; *Unverdorben et al., 2014*; *Estrin et al., 2013*). Our structure shows that Rpn8 and Rpn11 are the only subunits that contribute multiple helices to the bundle and together contact all other subunits within the helical assembly. Notably, Rpn8 is the largest contributor to the bundle, which is consistent with previous biochemical work showing that the Rpn8 C-terminal helices are more critical for lid assembly than those of other subunits (*Estrin et al., 2013*). The PCI horseshoe and MPN heterodimer are individually tethered to the bundle via short loops, but make only few direct surface contacts with the bundle. The cryoEM map also shows that the bundle's position in the isolated lid is likely stabilized by interactions between α-helix 5 (residues 186–215) of Rpn8 and the PCI domain of Rpn9 at one end of the bundle, and between the C-terminus of Rpn6 and the N-terminus of Rpn3 at the opposite end (*Figure 1E*, *Figure 1—figure supplement 7*).

The cryoEM reconstruction of the isolated lid allowed us to examine the structural elements involved in regulating Rpn11 DUB activity. Notably, within the isolated lid, the Rpn11/Rpn8 heterodimer is positioned in a previously unobserved orientation relative to the other subunits, stably associated within the palm of the hand-shaped complex via two primary interfaces with the **t**etratrico**p**eptide **r**epeat (TPR) domain of Rpn5 and the α-solenoid of Rpn9 (*Figure 2A–C*). The resulting organization produces the basis for Rpn11 DUB inhibition in the isolated lid.

## Rpn5 occludes the Rpn11 active site

We first probed the contacts between Rpn11 and Rpn5 for their contributions to Rpn11 inhibition in the isolated lid, as this interface is more extensive (total buried surface area of ~630 Å$^2$) than all other subunit interactions with the MPN heterodimer (*Figure 2—figure supplement 1*). Importantly, the N-terminal region of α-helix 13 in Rpn5 (residues 275–285) is nestled against the end of Rpn11's catalytic groove, with several residues from Rpn5 directly contacting loops that surround Rpn11's catalytic Zn$^{2+}$ ion (*Figure 2C*). To test the functional importance of these contacts, we generated Rpn5-mutated lid variants and compared their ubiquitin-7-amino-4-methylcoumarin (Ub-AMC) cleavage rate with that of wild-type lid and the isolated Rpn11/Rpn8 dimer. Under our assay conditions, Rpn11 activity within the isolated lid is 5-fold lower compared to the free Rpn11/Rpn8 dimer. In the loop preceding α-helix 13 of Rpn5, Tyr273 is in an orientation that enables hydrophobic interactions with Rpn11 Phe114, located in a loop near the active site (*Figure 2C*). Mutation of this Rpn5 residue (Y273A) increased Rpn11 activity to 61% of the isolated Rpn11/Rpn8 dimer (*Figure 2D–E*), suggesting that Tyr273 aids in stabilizing Rpn11 in its inhibited conformation.

Rpn5 residues His282 and Lys283 directly interact with the backbone atoms of two loops near the Rpn11 active site, and their substitution with alanine increased Rpn11 activity to 31% and 41% of the free MPN heterodimer, respectively (*Figure 2C, E*). The effects of these mutations were additive, as the Rpn5 (H282A,K283A) double-mutant lid exhibited 51% DUB activity compared to the free MPN

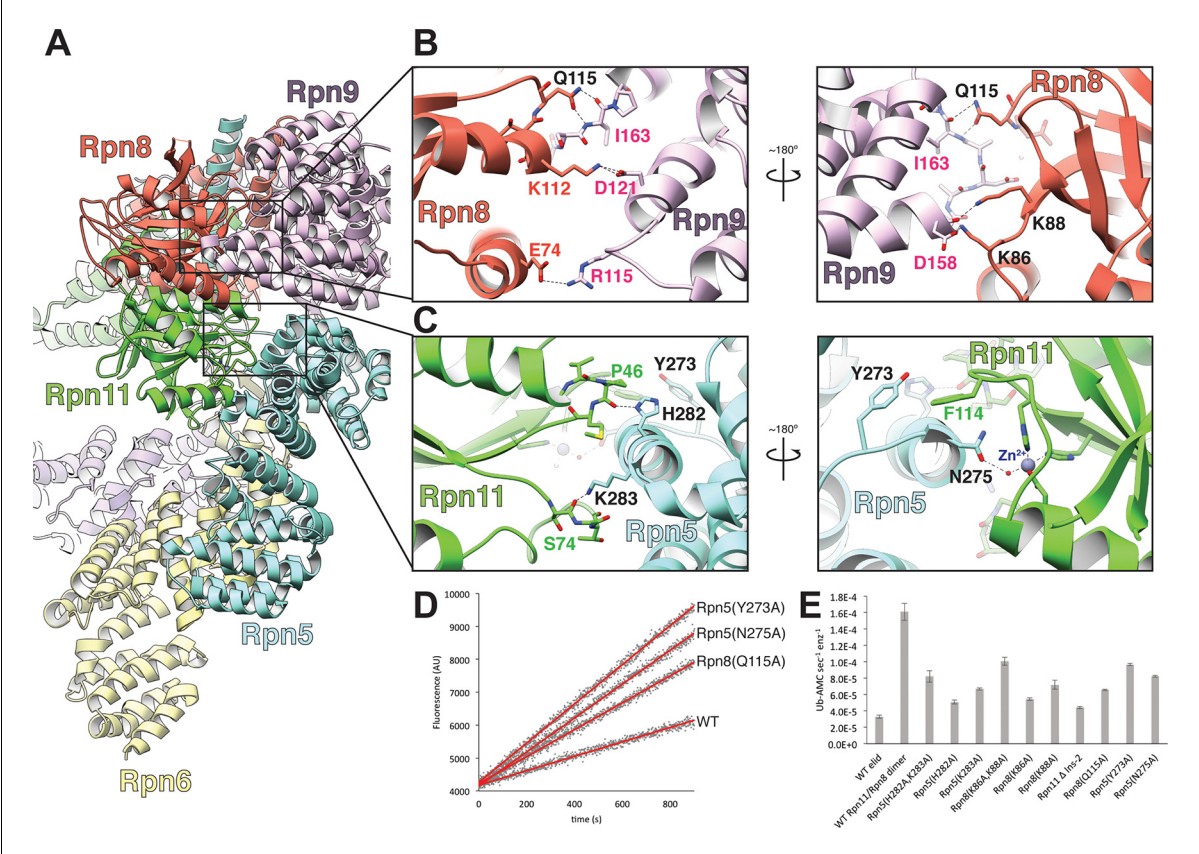

**Figure 2.** The MPN heterodimer interacts extensively with Rpn5 and Rpn9. (**A**) Side view of the lid sub-complex shows that the MPN heterodimer (Rpn8 in red, Rpn11 in green) interacts closely with the Rpn5 (blue) and Rpn9 (lavender) subunits. Side-chain interactions likely responsible for maintaining the MPN heterodimer in this configuration are shown in detail in panels (**B**) and (**C**). Residues that were mutated to alanine for deubiquitination assays are labeled in black. (**D**) Measurements of fluorescence increase upon Rpn11-mediated cleavage of ubiquitin-AMC are shown for three lid mutants relative to the wild-type lid. (**E**) Ubiquitin-AMC cleavage rates show activation of Rpn11 in the lid upon mutation of residues within Rpn5 and Rpn8.

The following figure supplements are available for figure 2:

**Figure supplement 1.** Buried surface area between the MPN heterodimer and Rpn9 and Rpn5.

**Figure supplement 2.** Interface mutations in Rpn5 and Rpn9 release the MPN dimer from its inhibited conformation.

heterodimer (*Figure 2E*). Structural analysis of the lid containing the Rpn5 (Y273A) or the Rpn5 (H282A,K283A) double-mutant by negative-stain EM shows that the Rpn11/Rpn8 heterodimer is released from its inhibitory conformation, while the overall organization of the PCI-containing sub-units is identical to that of the isolated wild-type lid complex (*Figure 2—figure supplement 2*). Together, these activating mutations support a model wherein Tyr273, His282, and Lys283 of Rpn5 all stabilize the association of α-helix 13 with the Rpn11 active site, generating a structural barrier that blocks substrates from accessing the catalytic groove.

In addition to preventing access to the Rpn11 active site by steric occlusion, the close proximity of Rpn5 in the isolated lid further blocks DUB activity through interaction with the catalytic $Zn^{2+}$ (*Figure 3A*). Two histidines (His109 and His111) and an aspartate (Asp112) coordinate the $Zn^{2+}$ within the Rpn11 active site, a configuration that is preserved in all JAMM metalloenzymes (*Komander et al., 2009*). This geometry allows for interaction with a fourth ligand, as $Zn^{2+}$ is usually tetrahedrally coordinated in proteins (*Gerke and Moss, 2002*). Despite the close proximity of Rpn5's α-helix 13 to the Rpn11 active site, intermolecular distances preclude direct interaction of any Rpn5 residues with the catalytic $Zn^{2+}$. The Rpn5 residue that is closest to the zinc is Asn275,

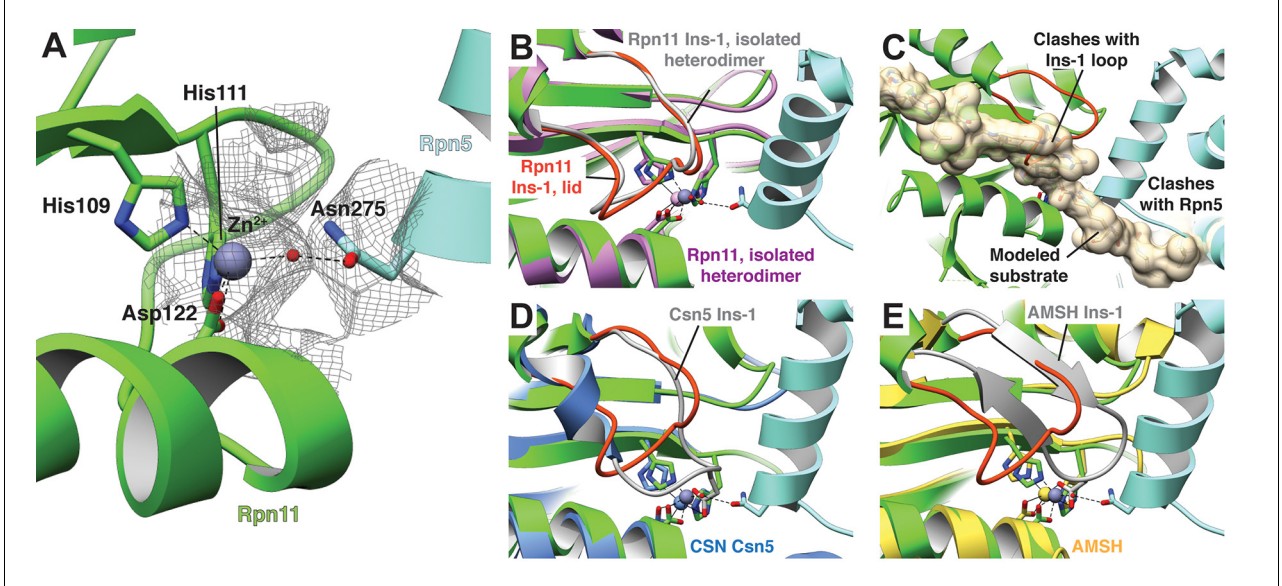

**Figure 3.** The Rpn11 active site is inhibited in the isolated lid. (**A**) The catalytic $Zn^{2+}$ (gray sphere) within the Rpn11 active site (green ribbon) is coordinated by three residues from Rpn11, and a water molecule acts as a fourth ligand, likely mediated by Asn275 from the neighboring Rpn5 subunit (blue). The cryoEM density in this region is shown as a mesh. (**B**) Comparison of the Rpn11 active sites from the isolated Rpn11/Rpn8 heterodimer crystal structure (PDB ID: 4O8X, purple) and isolated lid (green) shows that the two structures are nearly superimposable. (**C**) A di-ubiquitin substrate (tan) was modeled into the active site and shown as a transparent surface rendering. The modeled substrate severely clashes with the locked Rpn11 Ins-1 loop and Rpn5. (**D**) In CSN, a Glu within the Ins-1 loop provides a fourth point of coordination for the $Zn^{2+}$ ion. (**E**) Similar to CSN, an AMSH mutant utilizes an Asp from the Ins-1 loop to establish tetrahedral coordination of the catalytic $Zn^{2+}$.

The following figure supplements are available for figure 3:

**Figure supplement 1.** Water-mediated tetrahedral coordination of active site $Zn^{2+}$ in Rpn11 and AMSH orthologue Sst2.

**Figure supplement 2.** Relative B-values of the Rpn11 Ins-1 loop in the isolated Rpn11/Rpn8 heterodimer and in the isolated lid sub-complex.

which is notably oriented with its carboxamide group directed towards the Rpn11 active site. Mutation of Asn275 to alanine increases Rpn11 DUB activity in the isolated lid to 51% of the isolated MPN heterodimer (*Figure 2E*), and negative-stain EM of this lid mutant shows the Rpn11/Rpn8 heterodimer detached from its inhibited conformation (*Figure 2—figure supplement 2*).

Although Rpn5 Asn275 is not within range to directly bind the catalytic $Zn^{2+}$ (~5 Å from the $Zn^{2+}$ to Nδ1 of Asn275), the cryo-EM density in the Rpn11 active site shows connectivity between Asn275 and the catalytic $Zn^{2+}$ (*Figure 3A*), potentially corresponding to a coordinated water molecule. Indeed, a Zn-associated water molecule is known to play a key role in the peptide hydrolysis mechanism of Zn-dependent proteases and has been observed in the crystal structures of Rpn11 (*Worden et al., 2014*; *Pathare et al., 2014*) and related DUBs of the JAMM family, such as AMSH (*Shrestha et al., 2014*; *Davies et al., 2011*). Furthermore, the co-crystal structure of the AMSH ortholog Sst2 bound to a post-cleavage ubiquitin fragment shows that the carboxylate of ubiquitin Gly76 forms a hydrogen bond with the catalytic water (*Shrestha et al., 2014*) in the same manner as Rpn5 Asn275 in the isolated lid (*Figure 3—figure supplement 1*). While the Sst2 structure presents a snapshot of the transient substrate cleavage product prior to its departure from the active site, the positioning of Rpn5 Asn275 establishes a stable tetrahedral coordination of the $Zn^{2+}$ ion via this catalytic water molecule, inhibiting isopeptidase activity of Rpn11 in the isolated lid sub-complex.

## Rpn9 stabilizes the inhibited MPN heterodimer

The other major interface involved in stabilizing the DUB-inhibited conformation of the isolated lid is found between Rpn8 and Rpn9, and involves a 5-residue loop connecting α-helix 8 (residues 143–

159) and α-helix 9 (residues 165–182) of Rpn9. While the buried surface area of this interface (~450 Å²) is smaller than the Rpn5-Rpn11 interface (*Figure 2—figure supplement 1*), mutagenesis of the interface residues shows that these contacts also contribute significantly to maintaining the sequestered position of the MPN heterodimer within the palm of the isolated lid sub-complex.

Our atomic model suggests that Rpn8 Gln115 interacts with the backbone atoms of Rpn9 Ile163 (*Figure 2B*), and upon mutation of Gln115 to alanine, we observed elevated Rpn11 activity that was 33% of isolated MPN levels. Furthermore, two lysine residues in Rpn8, Lys86 and Lys88, are likely involved in electrostatic interactions with Rpn9 Asp158, which is located at the C-terminal end of α-helix 8 (*Figure 2B*). Mutation of Lys86 and Lys88 to alanine in Rpn8 of the isolated lid increases Rpn11 activity to 33% and 45% of the free MPN-heterodimer levels, respectively. The double mutant Rpn8 (K86A,K88A) was additive, stimulating Rpn11 activity to about 60% of the isolated MPN heterodimer. As with the Rpn5 (H282A,K283A) double mutant lid, negative-stain analysis of the Rpn8 (K86A,K88A) double mutant revealed that disruption of the Rpn8-Rpn9 interface releases the MPN dimer from its inhibited conformation (*Figure 2—figure supplement 2*).

Combined with the structural data, our mutational analyses of the MPN-dimer contacts with Rpn5 and Rpn9 suggest that DUB inhibition requires establishment of a finely tuned network of interactions and perturbation of this system at any of the identified contact points disrupts the inhibitory conformation of the MPN dimer within the isolated lid sub-complex.

## The Ins-1 loop blocks the active site

Common structural motifs present in many members of the MPN family are the two insertion loops, Ins-1 and Ins-2, which have been suggested to be involved in orienting ubiquitin chains for cleavage (*Sato et al., 2008*). In Rpn11, Ins-1 is required for catalysis and has been proposed to play a structural role in DUB activity by engaging and positioning the C-terminus of the ubiquitin substrate for hydrolysis (*Worden et al., 2014*). Flexibility of this loop suggests that it may regulate access to the DUB active site by switching between different conformational states. Upon ubiquitin binding to Rpn11, the Ins-1 loop may first open up to allow the ubiquitin C-terminus to enter the catalytic groove and then switch to a conformation that stabilizes the isopeptide bond for hydrolysis. Structures of the isolated Rpn11/Rpn8 dimer show the Ins-1 loop in a 'closed' conformation (*Worden et al., 2014*; *Pathare et al., 2014*), which is also observed in EM reconstructions of proteasomes that are actively processing a protein substrate (*Unverdorben et al., 2014*; *Matyskiela et al., 2013*). Interestingly, in the context of the isolated lid, the Ins-1 loop appears to be locked in this closed state through interactions with the neighboring Rpn5 subunit (*Figures 2B*, *3B*). In particular, the amino group of Rpn5 Lys283 interacts with the Ser74 carbonyl of Ins-1 (*Figure 2C*), and introducing the Rpn5 K283A mutation in the isolated lid results in a significant increase in Rpn11 DUB activity, as indicated above.

While the Ins-1 loop in the free Rpn11/Rpn8 heterodimer exhibited markedly elevated B-values (*Worden et al., 2014*; *Pathare et al., 2014*), the Rpn11 Ins-1 loop within the isolated lid has lower B-values than the average for all modeled Rpn11 atoms (*Figure 3—figure supplement 2*). These data suggest that the Ins-1 loop is locked in a closed conformation through contact with neighboring residues and is unable to switch to the 'open' state required for substrate access to the active site. The combined effects of the tetrahedral coordination of the catalytic $Zn^{2+}$ by Asn275 (*Figure 3A*), the steric hindrance imposed by Rpn5's α-helix 13 in the Rpn11 catalytic groove (*Figure 3C*), and the obstruction of the DUB active site by the Ins-1 loop result in robust DUB inhibition.

Interestingly, the proposed mechanism for auto-inhibition of the catalytically active MPN subunit in CSN, Csn5, also involves tetrahedral coordination of the active-site $Zn^{2+}$. However, the fourth ligand in Csn5 is not provided by a neighboring subunit, but intramolecularly by the Ins-1 loop that thereby gets stabilized in a closed conformation (*Lingaraju et al., 2014*) (*Figure 3D*). A similar scenario is also observed for a mutant AMSH construct (PDB ID: 3RZV) that utilizes a nearby Asp within the Ins-1 loop to complete the tetrahedral geometry (*Shrestha et al., 2014*) (*Figure 3E*).

## Incorporation of the lid into the 26S holoenzyme

Upon incorporation into the 26S proteasome, the lid undergoes major conformational changes that involve the PCI-assembly, the helical bundle, and especially the MPN heterodimer. To visualize these rearrangements, we compared the atomic coordinates of the isolated lid sub-complex to the

previously determined pseudo-atomic model of the lid in the context of the assembled proteasome (PDB ID: 4CR2) (*Unverdorben et al., 2014*) (*Figure 4* and *Video 1*). Lid binding to the base and core sub-complexes causes the PCI horseshoe to constrict, decreasing in radius by ~3 Å, and adopt a more planar conformation that closely resembles the reported architecture of the CSN (*Lingaraju et al., 2014*) (*Figure 4, Figure 1—figure supplement 6*). As a result, Rpn3, Rpn7 and Rpn12, comprising one half of the PCI horseshoe, undergo considerable rotation toward the center of the regulatory particle, where Rpn3 and Rpn12 bind the scaffolding subunit Rpn2, while Rpn7 contacts the AAA+ ATPase subunits Rpt3 and Rpt6. By comparison, the other half of the PCI horse-shoe, consisting of Rpn9, Rpn5 and Rpn6, goes through a much less pronounced conformational change. The N-terminal α-solenoid domain of Rpn9 extends toward the N-terminal coiled coil of Rpt4 and Rpt5, generating the binding site for the ubiquitin receptor Rpn10, and the highly flexible TPR segment of Rpn5 becomes stabilized through contact with the ATPase ring and the core pepti-dase. The N-terminal α-solenoid domain of Rpn6 also accommodates interactions with the ATPase ring and core peptidase by rotating ~34° around its long axis (*Figure 4—figure supplement 1*).

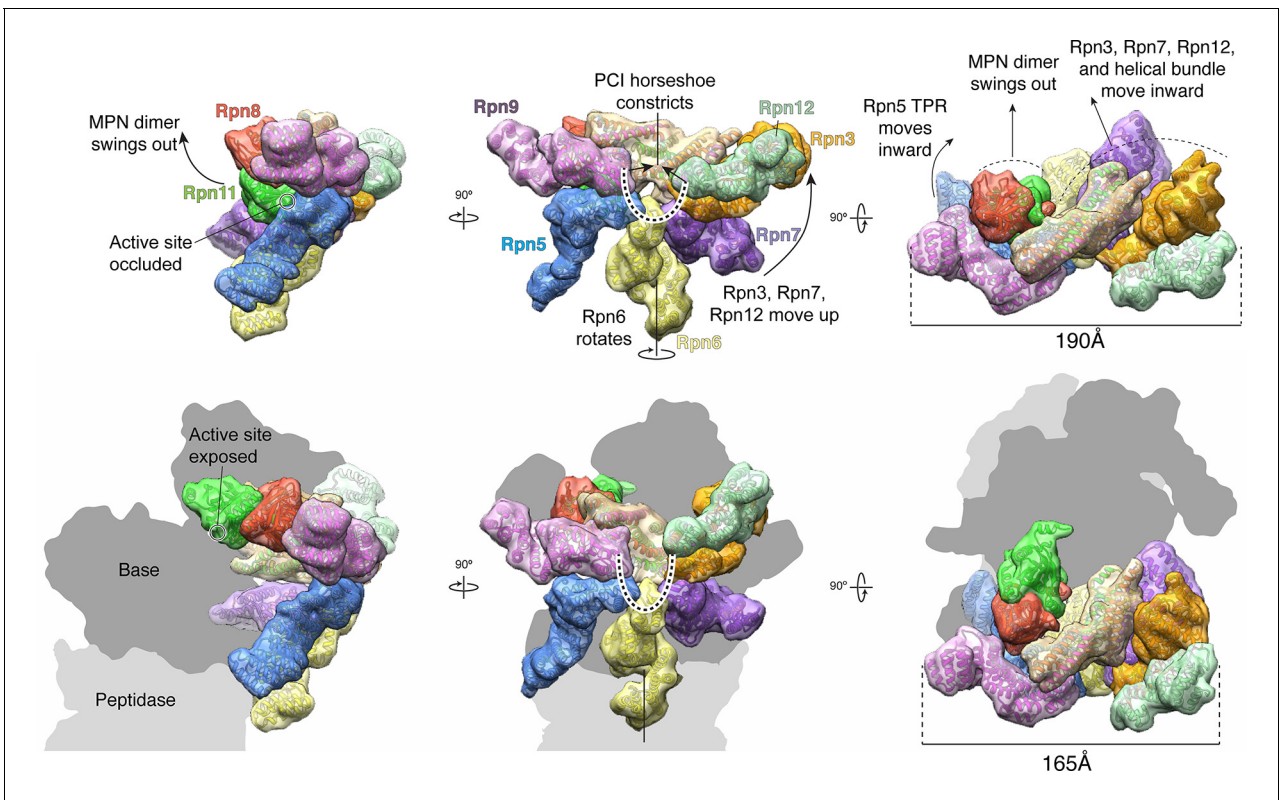

**Figure 4.** The lid sub-complex undergoes a dramatic reorganization upon incorporation to the 26S holoenzyme. Motions associated with lid incorporation are shown from three orthogonal views. Top panels correspond to the isolated lid, while bottom panels represent the proteasome-incorporated lid. Atomic models of the lid subunits were used to generate semi-transparent Gaussian filtered surfaces for visualization. For clarity, the helical bundle, which moves as a rigid body, is shown as a single surface. Sem1 is not shown. The base and core peptidase components are depicted as shadows to not occlude details of the lid rearrangement. Notable rearrangements include: a 90° rotation of the MPN dimer away from the inhibited conformation, movement of Rpn3, 7, and 12 away from Rpn5, 6, and 9, constriction of the PCI horseshoe, and an overall closure of the lid sub-complex around the regulatory particle.

The following figure supplements are available for figure 4:

**Figure supplement 1.** The Rpn6 α-solenoid domain rotates during incorporation into the holoenzyme.

**Figure supplement 2.** Rpn11's bundle helix binds to the N-ring of the holoenzyme ATPase.

**Figure supplement 3.** Lid incorporation activates Rpn11.

isolated lid

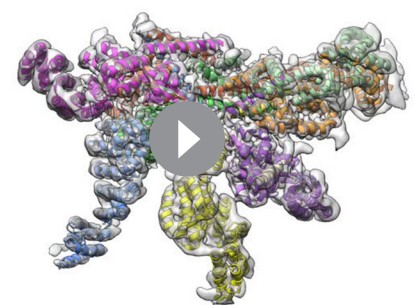

**Video 1.** Lid subunit rearrangements that occur during incorporation into the 26S holoenzyme. The atomic model of the isolated lid is interpolated into the pseudo-atomic model of the 26S holoenzyme (*Unverdorben et al., 2014*), showing the motions associated with lid incorporation.

The extensive rearrangements of the PCI-containing subunits upon interaction with base and core may also trigger movements of the helical bundle toward the ATPase ring of the base (*Figures 1A, B*, *2A*, and *4*). Because the bundle is connected to the PCI horseshoe through flexible loops, it can move as a single unit, ultimately adopting an orientation in the 26S holoenzyme that is more perpendicular to the hand-shaped arrangement of the PCI subunits. Both in the isolated and incorporated lid, the helical bundle contacts the N-terminal domain of Rpn3, albeit through different interfaces. That the association between these two components is maintained during lid incorporation suggests that movement of the Rpn3/7/12 unit influences the positioning of the bundle (*Figure 4* and *Video 1*). The final orientation of the helical bundle within the regulatory particle is likely also dictated by an extensive interaction between α-helix 6 of Rpn11 (residues 263–272) and the N-terminal domain ring of the ATPase subunits in the base (*Figure 4—figure supplement 2*).

The most pronounced conformational rearrangement of the lid involves the Rpn11/Rpn8 MPN-domain heterodimer. Upon lid incorporation, the Rpn11/Rpn8 dimer undergoes a dramatic 90°° rotation, moving from its inhibited state in the palm of the isolated lid to a highly extended conformation over the central substrate-translocation channel in the 26S holoenzyme (*Figure 4*). The inhibited conformation of the Rpn11/Rpn8 heterodimer in the isolated lid appears to be metastable, as mutations in either the Rpn5 or Rpn9 contact surfaces lead to release of the dimer. In fact, the extended conformation of the Rpn11/Rpn8 dimer in the proteasome is similar to its conformation in our DUB-activating lid mutants (*Figure 2—figure supplement 2*). During lid incorporation, it is likely that the conformational changes occurring in the PCI subunits upon their interactions with the core and base distort the Rpn11-Rpn5 and Rpn8-Rpn9 contact sites and release the Rpn11/Rpn8 dimer from its inhibited state. To assess the extent of Rpn11 activation upon lid incorporation, we compared Ub-AMC hydrolysis activity for Rpn11 in the isolated lid versus the assembled 26S proteasome. Incorporation of wild-type lid stimulated Rpn11 DUB activity to 150% of the isolated Rpn11/Rpn8 dimer levels. This hyperstimulation of Rpn11 in the proteasome may originate from an alternative Ins-1 loop conformation that is stabilized by the neighboring Rpt4/Rpt5 coiled coil. Rpn11 activity in lid sub-complexes that contain Rpn5 (Y273A), Rpn5 (N275A), or Rpn8 (K86A,K88A) mutations was also stimulated upon proteasome incorporation, although to a lower level than with the wild-type lid (*Figure 4—figure supplement 3*). None of the mutations are involved in interfaces between subunits in the proteasome holoenzyme, and we speculate that the slightly lower DUB activity of the reconstituted mutant proteasomes originates from an interference with normal lid incorporation due to a prematurely released MPN dimer. Interestingly, lid sub-complexes where Rpn11 lacked Ins-2 were completely deficient in Rpn11 stimulation upon incorporation into holoenzyme (*Figure 4—figure supplement 3C*), even though Rpn11 activity was unaffected by Ins-2 deletion in the isolated lid (*Figure 2E*). The Ins-2 region of Rpn11 is known to interact with the scaffolding subunit Rpn2 of the base and likely stabilizes the Rpn11/Rpn8 dimer in the extended conformation.

In summary, our structural and functional data suggest that during lid incorporation, the MPN-domain heterodimer loses its stabilizing interactions with Rpn5 and Rpn9, and extends out toward the center of the regulatory particle, leading to Rpn11 activation. This extended conformation enables the Rpn11 Ins-2 loop to interact with Rpn2, which likely aids in positioning the DUB active site above the entrance to the AAA+ ATPase ring for highly regulated deubiquitination of protein substrates during translocation.

## Conclusions

The primary function of the lid sub-complex is to house the isopeptidase Rpn11, an enzyme that is central to proteasomal substrate degradation. While earlier structural and biochemical work described the role of the lid scaffold in positioning Rpn11 and facilitating its activity in the context of the assembled proteasome holoenzyme, we illustrate here how interactions of the Rpn11/Rpn8 dimer with other lid subunits block premature DUB activity in the unincorporated lid assembly. Our atomic model of the isolated lid subcomplex showcases the dramatic conformational gymnastics undergone by this important component of the proteasome during incorporation into the regulatory particle and, while the molecular communication involved in promoting this massive reorganization is still an active area of structural and biochemical research, our work here has resolved an important mystery surrounding DUB inhibition and activation during proteasome assembly.

# Materials and methods

## Protein purification

Expression and purification of mutant and wild-type recombinant yeast proteasome lid complex was carried out in *E. coli* as described previously, with minor modifications (*Lander et al., 2012*). Briefly, *E. coli* BL21-star (DE3) cells containing the recombinant lid expression system (pETDuet-1 Rpn5, FLAG-Rpn6, Rpn8, Rpn9, 6xHis-Rpn11], pCOLADuet-1 [Rpn3, Rpn7, Rpn12] and pACYCDuet-1 [Sem1, Hsp90) were grown at 37°C in 6 liters of terrific broth (Novagen) supplemented with 150μM ZnCl$_2$. At OD$_{600}$ = 1.0, the temperature was reduced to 18°C and, at OD$_{600}$ = 1.5 lid, expression was induced overnight with 1 mM isopropyl-β-D-thiogalactopyranoside. After centrifugation, cell pellets were re-suspended in lid buffer (60 mM HEPES, pH8.0, 100 mM NaCl, 100 mM KCl, 10% Glycerol, 1 mM DTT) supplemented with protease inhibitors (Aprotinin, Pepstatin, Leupeptin, PMSF), 2mg/ml lysozyme, and bezonase. All purification steps were performed at 4°C. Cells were lysed by sonication and clarified by centrifugation at 16,000g for 30 min. Clarified lysate was incubated with anti-FLAG M2 resin (Sigma-Aldrich), washed with lid buffer and eluted with lid buffer supplemented with 0.15mg/ml 3x-FLAG peptide. FLAG eluate was concentrated to ~500 μl in a 30,000 MWCO spin concentrator (Amicon) and further purified by size-exclusion chromatography on a Superose 6 column (GE Healthcare) that was pre-equilibrated in lid buffer. Peak fractions were concentrated and stored at -80°C. Purification of core particle, Rpn10, Rpn11/Rpn8 MPN-domain dimer and recombinant base was performed as described previously (*Lander et al., 2012*; *Worden et al., 2014*; *Beckwith et al., 2013*).

## Rpn11 activity assay

All Ubiquitin-AMC cleavage experiments were performed at 30°C in lid buffer. Because Rpn11's Km for various ubiquitin substrates ranges from ~20 to ~300 μM, we assayed our WT and mutant lid variants at a constant, sub-Km Ubiquitin-AMC concentration. For all lid variants and the Rpn11/Rpn8 MPN-domain dimer, 500 nM enzyme was incubated with 2.5 μM Ubiquitin-AMC (Boston Biochem), and Rpn11-catalyzed ubiquitin cleavage was monitored by the increase in AMC fluorescence (Ex: 360 nm, Em: 435 nm) using a QuantaMaster spectrofluorometer (PTI). The slopes of individual time traces were translated to initial cleavage rates using a standard curve for ubiquitin-AMC (ranging from 0.5–2.5 μM) that had been completely cleaved by the DUB Yuh1. Ubiquitin-AMC cleavage rates for all variants were measured in triplicate except for WT lid, Rpn11/Rpn8 dimer, Rpn5 (H282A, K283A) and Rpn8 (Q115A), where n = 11, n = 6, n = 4, and n = 4, respectively.

## Rpn11 activation upon lid incorporation

Proteasomes were reconstituted in vitro with lid as the limiting component by mixing 250 nM lid, 375 nM core particle, 750 nM base and 1 μM Rpn10 in reconstitution buffer (60 mM HEPES, pH7.6, 100 mM NaCl, 100 mM KCl, 10% glycerol, 10 mM MgCl$_2$, 1 mM DTT, 0.5 mM ATP) that contained an ATP-regeneration system (5 mM ATP, 16 mM creatine phosphate, 6 μg/ml creatine phosphokinase). Deubiquitination reactions were initiated by the addition of 2.5 μM ubiquitin-AMC and monitored by the increase in AMC fluorescence (Ex: 360 nm, Em: 435 nm) using a QuantaMaster spectrofluorometer (PTI). A low level background DUB activity co-purified with our yeast core particle. To subtract this background activity, we reconstituted proteasomes as described above, but

with a lid variant containing Rpn11 active-site mutations that abolish zinc binding (Rpn11 [AxA]). The background DUB activity of Rpn11 (AxA) proteasomes was subtracted from the DUB activity of proteasomes reconstituted with WT Rpn11 to get the DUB activity that was specifically contributed by Rpn11. To directly compare the activity of proteasome-incorporated and unincorporated Rpn11, we monitored the ubiquitin-AMC hydrolysis activity of 250 nM lid and Rpn11/Rpn8 MPN-domain dimers in reconstitution buffer containing the ATP regeneration system but with core particle, base, and Rpn10 omitted.

## Electron microscopy sample preparation

For negative stain analysis, purified lid samples were diluted to ~50 nM in FLAG buffer (50 mM HEPES, pH7.6, 100 mM NaCl, 100 mM KCl) and directly applied to plasma-activated (20 s; 95% Ar, 5% $O_2$) copper grids for staining with 2% uranyl formate. For analysis by cryoEM, samples were diluted to ~5 uM in FLAG buffer that contained 1.5 mM TCEP (G Biosciences) and 0.05% NP-40 (Sigma). 4 µl of each sample was then applied directly to holey carbon C-flat grids (Protochips, 400 mesh, 1.2 µm holes) that had been plasma-cleaned (Gatan Solarus, 6 s; 95% Ar, 5% $O_2$) for manual blotting and plunge-freezing in liquid ethane.

## Electron microscopy

All imaging data was collected using automated Leginon imaging software (*Suloway et al., 2005*). Images of negatively stained samples of wild-type and mutant lid complexes were acquired on a Tecnai Spirit LaB$_6$ electron microscope operating at 120 keV, with a random defocus range of -0.5 µm to -1.5 µm and an electron dose of 20 e$^-$/Å$^2$. 331 images were acquired for wild-type lid, 433 images for the Rpn5 (H282A/K283A) double-mutant, 412 images for the Rpn8 (K86A/K88A) double mutant, 181 for the Rpn5 (N275A) mutant, and 204 for the Rpn5 (Y273A) mutant. Images were collected at a nominal magnification of 52,000 X on an F416 CMOS 4K X 4K camera (TVIPS) with a pixel size of 2.05 Å/pixel at the sample level.

Imaging of frozen hydrated samples was performed using a Titan Krios electron microscope operating at 300 keV, with a defocus range of -1.5 µm to -3.5 µm. A Gatan K2 Summit was used for counting individual electron events at a dose rate of 9.9e$^-$/pixel/s, using an exposure of 7.6 s consisting of 38 frames at 200 ms/frame. This resulted in a total electron dose of 43.8 e$^-$/Å$^2$, accounting for coincidence loss. A total of 3,432 images of wild-type lid were acquired at a nominal magnification of 22,500X, yielding a pixel size of 0.655 Å/pixel at the sample level when collected in super-resolution mode.

## Negative stain image processing

All image preprocessing was performed using the Appion image-processing pipeline (*Lander et al., 2009*). The contrast transfer function (CTF) was estimated using CTFFIND3 (*Mindell and Grigorieff, 2003*). For negative stain data, particles were selected using a difference of gaussians (DoG) picking algorithm (*Voss et al., 2009*), and only micrographs having an overall CTF confidence of greater than 80% were used for subsequent processing. The phases of the micrograph images were corrected according to the estimated CTF, and the particles were extracted using a box size of 160 pixels, and pixel values were capped at 4.5 sigma above or below the mean. Boxed particles were binned by a factor of 2 for processing. Reference-free 2D class averages of the extracted particles were determined through five rounds of iterative multivariate statistical analysis and multi-reference alignment (*Ogura et al., 2003*). The results of the 2D analysis were used to remove damaged, aggregated, or falsely selected particles from the dataset used for 3D analysis.

All 3D analysis was performed with RELION v1.31 (*Scheres, 2012*). Using a previously determined reconstruction of the wild type yeast proteasome lid as an initial model (EMD-1993) (*Lander et al., 2012*), a 3D refinement of 17,680 particles wild-type lid complex provided a reconstruction at 19.6 Å resolution, according to a Fourier Shell Correlation at 0.143 of two independently determined half-maps. This volume was used as the initial model for all 3D analysis of the mutant lid datasets. 3D classification was performed on each of the negative stain mutant lid datasets, and only 3D classes exhibiting well-ordered structural details were selected and combined within each dataset for 3D refinement. 22,103 particles of the Rpn5 (H282A/K283A) mutant yielded a 25.2 Å reconstruction; 11,185 particles of the Rpn8 (K86A/K88A) mutant yielded a 27.3 Å reconstruction; 25,429 particles

of the Rpn5 (N275A) mutant yielded a 23.4 Å reconstruction, and 44,272 particles of the Rpn5 (Y273A) mutant yielded a 21.8 Å reconstruction (*Figure 2—figure supplement 2*). UCSF Chimera (*Goddard et al., 2007*) was used to dock the atomic model model of the lid into the density.

## Cryo-EM image processing

For cryo-EM image preprocessing, the super-resolution images were binned by a factor of two in reciprocal space, and motion-corrected using MotionCorr (*Li et al., 2013*). The aligned frames were summed and used for all subsequent processing steps. The CTF was estimated using CTFFIND3 (*Mindell and Grigorieff, 2003*), and only micrographs having a CTF confidence value that was greater than 50% at 4Å resolution were used for further processing (*Figure 1—figure supplement 1C*), resulting in a dataset of 3,365 micrographs. Particles were manually selected from the first 100 images, and the results of reference-free 2D analysis were used as templates for particle selection using FindEM (*Roseman, 2004*). A random subset of 50,000 particles were extracted from the micrographs with a box size of 256 and used for reference-free 2D analysis in order to rapidly assess the quality of particle selection (*Figure 1—figure supplement 1B*). Very few classes corresponding to damaged or aggregated particles were observed; so all particles were used for single particle analysis in RELION.

A total of 254,112 particles were extracted from the micrographs using a box size of 288 pixels, binned by a factor of 4, and classified into 8 3D classes over the course of 22 iterations in RELION. The particles from the 4 classes that showed evidence of conformational and compositional stability were selected from this initial classification, providing a total of 139,561 particles. The x and y coordinates corresponding to these particles were adjusted according to the final translational alignments from the 3D classification, and the centered particle coordinates were used to extract an unbinned particle dataset for 3D refinement in RELION.

3D refinement using the default RELION parameters yielded a 4.4 Å resolution structure after 22 iterations. These aligned particle parameters were used for the RELION 'particle polishing' method. Individual particle motion trajectories were estimated using a running average window of 7 frames and particle translations were limited using a prior with a standard deviation of 1. Particle movements were fit to a linear trajectory using a running average window of 7 frames, with an inter-particle distance contribution value set to 300 pixels. Per-frame B-factors and intercepts were estimated by comparing the reconstructed half-maps from individual frames to the full-frame half maps, and the spatial frequency contribution from each frame weighted according (*Figure 1—figure supplement 1D, E*). A new stack of particles was generated from the translationally aligned particles extracted from the weighted frames, which provided a reconstruction at 4.1 Å resolution.

Due to the possibility that the flexible N-terminal domains of the PCI subunits were negatively influencing the particle alignment, a soft-edged 3D mask encompassing the PCI-domains, the helical bundle, and the MPN domains was generated (blue mask shown in *Figure 1—figure supplement 2*) and used for 3D classification of the particles into 3 classes. This 3D classification was performed using the alignments from the 3D refinement, without further alignment of particles. One of the 3D classes resulting from this analysis clearly exhibited higher resolution details than the other two, and the 109,396 particles contained in this class were further refined (in the absence of a mask) to achieve a 3.6 Å structure. The same soft-edged 3D mask that was used for the previous 3D classification was then used for continued 3D refinement, which improved the structural details of the region contained within this mask, and increased the resolution to 3.5 Å resolution.

## Modeling

Modeling and visualization of the lid was performed in COOT (*Emsley and Cowtan, 2004*) using mostly the cryo-EM map that had been generated using a soft mask encompassing the PCI domains and the C-terminal helical bundle (deposited as EMD-6479), as this is the highest resolved region, and cross-validated using the unmasked map. Available structures and homology models generated using Modeller v9.15 (*Eswar et al., 2007*) were initially fit into the unmasked cryo-EM map using Chimera (*Goddard et al., 2007*). These included: 1) the crystal structure of *Drosophila melanogaster* Rpn6 (residues 50–390) homolog (PDB ID: 3TXN) (*Pathare et al., 2014*); 2) the crystal structure of the *Saccharomyces cerevisiae* Rpn11-Rpn8 heterodimer (residues 24–220 and 10–280, respectively;

PDB ID: 4O8X) (*Worden et al., 2014*); 3) the NMR structures of the N-terminal (residues 4–140 (PDB ID: 2MQW) and C-terminal (residues 184–353 (PDB ID: 2MRI)) domains of *Saccharomyces cerevisiae* Rpn9 (*Hu et al., 2015*); and 4) the N-terminal domain of *Schizosaccharomyces pombe* Rpn12 homolog (residues 6–200, PDB ID: 4B0Z) (*Boehringer et al., 2012*). The most N-terminal helices of Rpn5 and Rpn6 were not modeled due to the limited resolution of these regions. Placement of the N-terminal helices of Rpn3 was possible, however the absolute sequence register could not be assigned and these helices were modeled as polyalanine.

Following each round of real space refinement in Phenix v1.10 (*Adams et al., 2010*), 100 models were generated in Rosetta (*DiMaio et al., 2015*), clustered, and scored. The top scoring structures were then used for the next round of manual model building and an aggregate model was used for refinement in Phenix. For the final round of refinement, the SHAKE protocol in Phenix was used to displace all atoms of the top 5 scoring models by 0.5 Å before refinement against one of the unmasked half-maps. An ensemble of these 5 models have been deposited in the PDB under ID: 3JCK.

## Visualizing rearrangements involved in lid incorporation to the 26S

To visualize conformational changes undergone by the lid complex upon incorporation into the 26S proteasome, we first rigid-body fit individual components of the atomic model of our isolated lid (6 PCI domains, 6 N-terminal extensions, the MPN heterodimer, and the helical bundle) onto the pseudo-atomic model of the engaged lid (PDB-ID: 4CR2) (*Unverdorben et al., 2014*) using the 'MatchMaker' tool in Chimera. These overlaid models were then docked into the EM density of the 26S holoenzyme in the S1 state (*Unverdorben et al., 2014*). Overall, the secondary structure organization of the atomic models matched with high fidelity, although the register of the C-terminal helices of Rpn11 and the N-terminal helices of Rpn9 of the incorporated lid model were modified to correspond to the isolated lid model. The domain movements were visualized using the 'morph conformations' tool in UCSF Chimera. The motion of Rpn6 was evaluated using the software DynDom (*Hayward and Berendsen, 1998*).

## Accession codes

The EM density maps for the 26S proteasome lid sub-complex before and after masked refinement, as well as unsharpened maps and half-maps, have been deposited in the Electron Microscopy Data Bank under accession number EMD-6479. The atomic coordinates of the five highest-scoring Rosetta models have been deposited under accession ID: 3JCK in the PDB.

## Acknowledgements

We thank the members of the Lander and Martin labs for helpful discussions, and particularly Saikat Chowdhury and Lyn'Al Nosaka from GCL's lab for their help with EM data collection and processing. We are also grateful to Jean-Christophe Ducom at TSRI HPC for establishing the necessary computational infrastructure to process the EM data. This research was funded in part by the Damon Runyon Cancer Research Foundation (DFS-#07-13), the Pew Scholars program, the Searle Scholars program, and the US National Institutes of Health (grant DP2 EB020402-01) to GCL. AM acknowledges support from the Searle Scholars Program, the US National Institutes of Health (grant R01-GM094497), the US National Science Foundation CAREER Program (NSF-MCB-1150288), and the Howard Hughes Medical Institute. EJW acknowledges support from the US National Science Foundation Graduate Research Fellowship.

## Additional information

### Funding

| Funder | Grant reference number | Author |
|---|---|---|
| Damon Runyon Cancer Research Foundation | DFS-#07-13 | Gabriel C Lander |
| Pew Charitable Trusts | | Gabriel C Lander |

| | | |
|---|---|---|
| Kinship Foundation | | Gabriel C Lander |
| National Institutes of Health | DP2 EB020402-01 | Gabriel C Lander |
| National Science Foundation | NSF-MCB-1150288 | Andreas Martin |
| National Institutes of Health | R01-GM094497 | Andreas Martin |

The funders had no role in study design, data collection and interpretation, or the decision to submit the work for publication.

### Author contributions

CMD, GCL, Performed the EM data collection and processing, Conception and design, Acquisition of data, Analysis and interpretation of data, Drafting or revising the article; EJW, Expressed and purified wild type and mutant proteasome components for EM analysis, and performed biochemical experiments, Conception and design, Acquisition of data, Analysis and interpretation of data, Drafting or revising the article; MAH, Performed the atomic modeling, Analysis and interpretation of data, Drafting or revising the article; AM, Conception and design, Acquisition of data, Analysis and interpretation of data, Drafting or revising the article

## Additional files

### Major datasets

The following datasets were generated:

| Author(s) | Year | Dataset title | Dataset URL | Database, license, and accessibility information |
|---|---|---|---|---|
| Herzik Jr MA, Dambacher CM, Worden EJ, Martin A, Lander GC | 2016 | Structure of the yeast 26S proteasome lid sub-complex | http://www.rcsb.org/pdb/explore/explore.do?structureId=3JCK | Publicly available at the RSCB Protein Data Bank (accession no. 3JCK) |

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
