## [Decision Letter]

Thank you for submitting your work entitled "Atomic structure of the 26S proteasome lid shows the mechanism of deubiquitinase inhibition" for consideration by *eLife*. Your article has been favorably evaluated by Ivan Dikic (Senior editor) and three reviewers, one of whom, Sjors Scheres, is a member of our Board of Reviewing Editors. The following individual involved in review of your submission has agreed to reveal their identity: David Barford (peer reviewer). A further reviewer remains anonymous.

The reviewers have discussed the reviews with one another and the Reviewing Editor has drafted this decision to help you prepare a revised submission.

Summary:

This paper describes a near-atomic resolution cryo-EM structure of the free lid sub-complex from the 26S proteasome. The 26 S proteasome mediates the ubiquitin-dependent proteolysis of many regulatory and unfolded proteins. It plays essential roles in virtually all biological processes. Thus structural and mechanistic studies of this large and highly complex macromolecular assembly are extremely important. The main interest of the free lid sub-complex is to see how it compares to the structure of the lid in the context of the mature 26S proteasome, which is known from previous near-atomic resolution cryo-EM structures. Most significantly, the structure helps to rationalize why the Rpn11 ubiquitin isopeptidase is nearly inactive in the free lid as a result of conformational and steric obstruction of the Rpn11 active site, especially by the N-terminal domain of Rpn5. This is supported by mutational and enzymatic activity analyses.

The paper is well written and deemed appropriate for publication in *eLife*, provided the essential revisions below are addressed adequately.

Essential revisions:

It would be useful to mention if the crosslinking analysis recently done on the lid, as reported in Cell 163:432-44, is consistent with the cryo-EM and mutational data in the current work. It should at least be cited as a previous method used to analyze lid structure and its changes upon engagement with the full proteasome since this earlier report also indicated large conformational changes between free and engaged lid.

At 3.5 Angstrom resolution one cannot see individual water molecules! Therefore, the discussion in the subsection “Rpn5 occludes the Rpn11 active site” regarding the water molecule should be carefully reworded. The other crystal structures of Zn-associated water molecules can provide suggestions that water is also present in this structure, but the current experimental data do *not* indicate that water is present in this structure.

---

## [Author Response]

*This paper describes a near-atomic resolution cryo-EM structure of the free lid sub-complex from the 26S proteasome. The 26 S proteasome mediates the ubiquitin-dependent proteolysis of many regulatory and unfolded proteins. It plays essential roles in virtually all biological processes. Thus structural and mechanistic studies of this large and highly complex macromolecular assembly are extremely important. The main interest of the free lid sub-complex is to see how it compares to the structure of the lid in the context of the mature 26S proteasome, which is known from previous near-atomic resolution cryo-EM structures. Most significantly, the structure helps to rationalize why the Rpn11 ubiquitin isopeptidase is nearly inactive in the free lid as a result of conformational and steric obstruction of the Rpn11 active site, especially by the N-terminal domain of Rpn5. This is supported by mutational and enzymatic activity analyses. The paper is well written and deemed appropriate for publication in* eLife

*, provided the essential revisions below are addressed adequately.*

We thank the reviewers for their assessment of the manuscript. We feel obliged, however, to point out that the mature 26S proteasome is *not* known to near-atomic resolution. Currently, the highest resolution structure of the 26S proteasome is around 7 Å, which is far from near-atomic.

We have made the requested changes to the manuscript, as explained below.

*Essential revisions: It would be useful to mention if the crosslinking analysis recently done on the lid, as reported in Cell 163:432-44, is consistent with the cryo-EM and mutational data in the current work. It should at least be cited as a previous method used to analyze lid structure and its changes upon engagement with the full proteasome since this earlier report also indicated large conformational changes between free and engaged lid.*

We discuss the results of the crosslinking study in light of our work in the first paragraph of the “Lid architecture” section.

At 3.5 Angstrom resolution one cannot see individual water molecules! Therefore, the discussion in the subsection “Rpn5 occludes the Rpn11 active site” regarding the water molecule should be carefully reworded. The other crystal structures of Zn-associated water molecules can provide suggestions that water is also present in this structure, but the current experimental data do not indicate that water is present in this structure.

Upon re-reading this section of the manuscript, we agree that we overstated our ability to interpret the electron density as a water molecule. However, this region of the map is closer to 3Å resolution, and although we don’t observe a well-defined punctate density corresponding to a water molecule, we do see an extension of the EM density between the Zn and the Asn275, and we think it’s appropriate to mention that the density is consistent with the previously observed Zn-associated water molecule in crystallographic studies. We have, as suggested, substantially softened the language regarding this density (subsection “Rpn5 occludes the Rpn11 active site”, last paragraph).